# Colorectal Cancer in Young and Older Adults in Uruguay: Changes in Recent Incidence and Mortality Trends

**DOI:** 10.3390/ijerph18158232

**Published:** 2021-08-03

**Authors:** Carina Musetti, Mariela Garau, Rafael Alonso, Marion Piñeros, Isabelle Soerjomataram, Enrique Barrios

**Affiliations:** 1Registro Nacional de Cáncer Uruguay, Montevideo CP 11200, Uruguay; garaum@urucan.org.uy (M.G.); alonosor@urucan.org.uy (R.A.); ebarrios@urucan.org.uy (E.B.); 2Departamento de Métodos Cuantitativos, Facultad de Medicina, Universidad de la República, Montevideo CP 1180, Uruguay; 3Cancer Surveillance Branch, International Agency for Research on Cancer, CP 69372 Lyon, France; pinerosm@iarc.fr (M.P.); SoerjomataramI@iarc.fr (I.S.)

**Keywords:** colorectal cancer, incidence, mortality, trends, Uruguay, registry, national

## Abstract

Uruguay has the highest colorectal cancer incidence rates in Latin America. Previous studies reported a stable incidence and a slight increase in mortality among males. We aimed to assess colorectal cancer incidence (2002–2017) and mortality trends (1990–2017) by age groups and sex, using data from the National Cancer Registry. Annual percent changes (APCs) were estimated using joinpoint regression models. We included 27,561 colorectal cancer cases and 25,403 deaths. We found an increasing incidence among both males and females aged 40–49, with annual increases of 3.1% (95%CI: 1.21–5.03) and 2.1% (95%CI: 0.49–3.66), respectively, and an increasein the rate in older males (70+) of 0.60% (95%CI: 0.02–1.20) per year between 2002 and 2017. Mortality remained stable among those younger than 50, whereas it decreased for older females aged 50–69 and 70+ (APC: −0.61% (−1.07–0.14) and −0.68% (−1.02–0.34), respectively), and increased for the oldest males (70+; APC: 0.74 (0.47–1.01)). In conclusion, we found rising colorectal cancer incidence accompanied by stable mortality in young adults. Sex disparities were also found among the older adults, with a more favorable pattern for females. Exposures to dietary and lifestyle risk factors, and inequalities in access to and awareness of screening programs, are probably among the main underlying causes and deserve further investigation.

## 1. Introduction

Colorectal cancer (CRC) is one of the most common cancers diagnosed globally, ranking fourth after breast, prostate, and lung cancer for both sexes combined. Although CRC is a major health problem for males and females, there are substantial sex differences in incidence and mortality rates globally. In 2020, 1,065,960 males (Age Standardized Rate (ASR) 23.4/100,000) and 865,630 females (ASR 16.2/100,000) developed this disease, and 515,637 males (ASR 11/100,000) and 419,536 females (ASR 7.2), respectively, died from this cancer globally. Overall, the estimated cumulative risk of developing colorectal cancer is about 2.7% for men and 1.8% for women aged 0 to 74 [1].

CRC risk increases with age. Almost 60% of the new cases occur in people older than 65 and, conversely, less than 10% of the cases are diagnosed in people younger than 50. Differences in incidence and mortality by sex become more evident in older adults [1]. The available evidence supports the association of dietary patterns characterized by a high intake of red and processed meat, sugar, and refined grains, and low fiber consumption, with a higher risk of CRC. There is convincing evidence that overweight and obesity, insufficient physical activity, alcohol consumption, and smoking increase the risk of developing CRC [2,3]. Given the association between lifestyle and the risk for developing this cancer, it is likely that sociocultural and gender components, and not only biological factors, contribute to the observed differences between males and females [4,5,6].

CRC incidence rates show important variation across countries and are strongly correlated to the Human Development Index (HDI), with the highest incidence rates found in higher HDI settings. Almost 50% of the cases globally are diagnosed in countries indexed with a very high HDI [1]. Incidence and mortality time trends also display a correlation with the HDI, according to Arnold et al. In many medium- and low-HDI countries that are currently undergoing general improvements in life conditions, urbanization, and adoption of Westernized lifestyles, CRC incidence and mortality have shown increasing trends, whereas rates have decreased in countries assigned the highest levels of development [7]. For each analyzed country, the authors found that trends were similar for both sexes, despite differences in rates.

Uruguay, with a population of 3.5 million, of which more than 95% is urbanized [8], has evolved from a high- to a very high-HDI country [9]. It has the highest CRC incidence rates among Latin American countries, and ranks in the highest quintile globally [1,10] (ASR 38/100,000 males and 28/100,000 females, 2012–2016) [11]. A previous analysis including cases and deaths from CRC between 2002 and 2015 showed stable CRC incidence in both sexes [12], whereas mortality declined in females with an annual percent change (APC) of −0.5 and increased in males (APC, +0.3). Both changes are mild but significant [11]. The intention of the present study was to revisit these differences in trends by sex, through age group analysis.

In Europe, North America, and Australia, studies have shown an increase in incidence rates among young adults and a shift towards a younger age of onset, changes that may be related to lifestyle changes [13,14,15,16,17,18,19]. In turn, decreases in mortality among those aged 50 years and older have been attributed to screening [20]. In Uruguay, opportunistic CRC screening for people aged 50 to 74 years has been ongoing since the mid-1990s. The current guideline recommends an immunochemical fecal occult blood test (iFOBT) every two years, followed up with colonoscopy for positive tests [21,22,23]. If this program were successful, a decrease in invasive CRC incidence rates would be expected in the older population reflecting the detection of premalignant lesions (polyps).

In this study, we aimed to perform a time-trend analysis by age group and sex to gain an understanding of the present situation with regards to colorectal cancer in Uruguay. We were mainly interested in assessing the impact of the public health measures implemented thus far, with a special focus on trends among people below the screening age and sex differences that may lead to new insights.

## 2. Materials and Methods

### 2.1. Data Sources

The National Cancer Registry of Uruguay (NCRU) is a population-based cancer registry (PBCR) that has recorded information at the national level since 1991 [12]. It collects, stores, analyzes, and disseminates high quality cancer data [24,25,26]. All cases of invasive CRC (ICD-O-3 code C18-C20) diagnosed from 2002 to 2017 were extracted from the NCRU DataBase (DB) and included in the analysis. Carcinoid tumor from the appendix (ICD-O-3 code C18.1, morphology code 8240/3) was excluded.

The NCRU also records cancer deaths based on information directly abstracted from all death certificates in the country. CRC deaths from 1990 to 2017 were also extracted from the DB for this study. Both incidence and mortality cases were coded using ICD-O-3 (cases recorded prior to 2005 that were originally coded with version 2 were recoded to version 3) [27].

### 2.2. Data Analysis

Incidence and mortality rates (expressed per 100,000 person-years) were age adjusted using the direct method and the World Standard Population [28]. Age-specific rates were analyzed for four separate age groups (20–39, 40–49, 50–69, and 70+). Person-years at risk were estimated using the country’s population by linear interpolation of census data from 1996, 2004, and 2011 [8,29,30].

Time trends were assessed using joinpoint regression to detect points in time at which changes occur, by fitting a series of joined straight lines on a log-scale to age-adjusted rates over time. The publicly available software, Joinpoint version 4.7, from the Surveillance Research Program of the US National Cancer Institute, was used [31,32]. A maximum of two joinpoints were allowed. Trends were reported as increasing or decreasing when the annual percent change (APC) was statistically significant (*p* < 0.05); otherwise, the trend was described as stable (flat). For each APC estimate, the corresponding 95% confidence interval (95% CI) was also calculated.

The data from the National Cancer Registry Database were collected under strict personal data protection rules for epidemiological surveillance and therefore approval from the Ethics Committee was not required.

## 3. Results

Between 2002 and 2017, 27,561 incident colorectal cancer cases were registered by the NCRU, representing about 14% of all registered cancers in Uruguay [33]. The median age at diagnosis was 73 years for females and 71 years for males, which remained stable across the whole study period. The number of cases and proportion by age group according to sex are presented in Table 1. Although less than 7% of new cases were diagnosed in people under 50, more than 50% were diagnosed in people aged 70+.

### 3.1. Incidence Time Trends 2002–2017

Between 2002 and 2017, CRC incidence rates slightly increased among males (APC: 0.79) and remained stable in females (Table 2, Figure 1). The age-specific analysis showed an important increase in incidence rates for both sexes among people aged 40–49, with annual increases of 3.1% and 2.1% for males and females, respectively. Rates remained stable for both sexes in the youngest (20–39) and middle-aged (50–69) groups, and displayed a minor increase among older males (70+, APC: 0.6%). In this latter age group, incidence rates remained stable among females. The annual percent change (APC) and the 95% confidence interval are presented for each sex and age group in Table 2, and time trends are displayed in Figure 1 and Figure 2.

### 3.2. Mortality Time Trends 1990–2017

A total of 25,403 deaths were analyzed over a 28 year period. CRC mortality increased in males by 0.28% annually and decreased in females by −0.66% (Table 3, Figure 1). For males, an increase in mortality rates was observed, yet only among people older than 70 (APC: 0.74%). Regarding females, significant decreases were observed in those aged 50 to 69 and older than 70 (APC: −0.61% and −0.68%, respectively). It is noteworthy that for people younger than 50, mortality remained stable in both sexes (Table 3).

## 4. Discussion

Our study was based on national data from a population-based cancer registry with high data quality [24,25,26]. It is, to our knowledge, the first locally developed time-trend analysis for CRC incidence and mortality by sex and age group published in any Latin American country. For all ages combined, we confirmed sex differences in time trends, as previously reported [12]. Although Uruguay has had a very high HDI status during the past two decades [9], the country does not fit into any of the patterns described by Arnold et al. [7]. For example, for females, observed trends in incidence and mortality appear to correspond to a higher HDI category than those for males. The following discussion focuses on two key findings: the increasing incidence among young adults and the divergent trends by sex in older adults.

### 4.1. Increasing Incidence among Young Adults

Similar to other very high HDI countries [13,14,15,16,17], we also observed increasing incidence rates among young adults (age 40–49), with stable mortality rates in Uruguay. Other studies performed over longer periods of observation and including age-period-cohort analysis have attributed this increase to a cohort effect that is related to several dietary and lifestyle changes, especially among the youngest cohorts [15,16,17,18]. Obesity has been associated with increasing CRC incidence in all age groups [34,35] and its prevalence has rapidly risen, particularly among young adults. In Uruguay, the prevalence of overweight and obesity (Body Mass Index > 25 kg/m^2^) increased from 56.6% to 64.9% between 2006 and 2013 among people aged 25 to 64 [36], and this may explain part of the observed rise in CRC incidence. Furthermore, high consumption of processed foods and high-glycemic load carbohydrates—which is a global phenomenon [37]—have been reported to create an inflammatory bowel environment that may lead to the proliferation of colonic cells, and may therefore increase CRC risk [35,38]. Nonetheless, although the relative contribution of dietary risk factors is well established for older adults, their role in the early onset of CRC remains unclear.

Although the vast majority of CRCs are sporadic, hereditary syndromes (HS) should be considered in early onset CRC, especially among cases diagnosed before the age of 40. The two main forms of HS for CRC are Hereditary Nonpolyposis CRC (HNPCC), also known as Lynch’s syndrome, and Family Adenomatous Polyposis (FAP) [39]. A retrospective study in the US reported that some germ line mutations associated with HS were detected in almost one out of five CRC cases diagnosed in people younger than 50 years [40]. In Uruguay, a single center study estimated that 2.6% of cases diagnosed fulfilled clinical criteria for HNPCC, and an additional 5.6% could be considered as a population at increased risk. When only individuals below age 50 were considered, the proportion increased to 10.8% [41]. In more recent years, the same researchers found that among individuals who fulfilled clinical criteria for HNPCC, almost one-quarter presented germ line mutations [42]. Although the rapid rises in CRC incidence observed in Uruguay are not likely caused by changing genetic profiles within the population, these increases call for specific health services targeted at those families affected by hereditary syndromes, or those individuals considered to present increased risk, including genetic testing, counseling, and preventive interventions (tailored endoscopic follow up, Cox2 inhibitors, and prophylactic surgery). Furthermore, because most early onset CRCs are diagnosed in people without a clear family history, the promotion of a healthier lifestyle, upstream policy measures to ensure healthy life choices, increased awareness among patients and the medical community about CRCs among younger patients, and the discussion of key aspects in screening policies, such as the lower age limit, are of utmost relevance.

Another contributing factor to increased CRC diagnosis among the younger population in Uruguay may be the broad availability of screening tests that may have led physicians to offer iFOBT to healthy adults in their forties, although not formally recommended. This practice could cause some degree of “advancement” in diagnosis, leading to an earlier stage of disease and better cure rates. It is also recognized that stage-specific survival and cure rates are better for younger than for older adults, a factor that could contribute to the stable mortality among this age group [43].

### 4.2. Divergent Trends by Sex in Older Adults

Overall, the findings reported here mostly reflect the cancer burden and risks for the older population age group (70+), which accounts for over 55% of cases.

Unlike other countries in the very high HDI group, we found diverging trends by sex among older adults (70+), with increasing incidence and mortality among males, and stable incidence with decreasing mortality in females. These differences could be explained by contrasting exposure to established CRC risk factors between sexes or diverging attitudes towards health care, in particular regarding participation in screening programs.

In Uruguay, dietary intake of fruits and vegetables is low for over 90% of the population, almost 60% of the population is overweight or obese, alcohol and tobacco consumption are high, and physical activity is insufficient [36]. Furthermore, an increase in the consumption of processed meat has occurred in recent decades. According to the Foreign Agricultural Service of the United States Department of Agriculture (FAS-USDA), Uruguay consistently ranks as one of the top per capita beef consumers in the world, behind Argentina and USA [44]. Although there are clear differences in the prevalence of alcohol and tobacco consumption among sexes, which is much higher among males, disparities in dietary patterns and physical activity are less evident. Furthermore, no major differences in overweight or obesity were observed [36].

Another key factor to be explored is the influence of participation in screening programs, and its impact on mortality trends. In Uruguay, adherence to colorectal cancer screening is heterogeneous and, until recent years, improving the coverage was challenging [22,23]. To avoid differences in implementation across the country, a National Comprehensive Guideline for colorectal cancer screening, available in Spanish, was launched in 2018. Although this represents a major improvement, no sex-specific recommendations are considered [21]. According to the 2013 National Noncommunicable Disease Risk Factor Survey, only 36% of males older than 50 had at least one iFOBT, compared to 46.3% of females [36]. A lower participation of males in CRC screening programs was previously reported by several qualitative investigations, summarized by Shannon Christy (et al.). They confirmed the presence of cultural and social barriers that prevent males (and particularly Latinos) from participating in CRC screening programs in USA [45]. Some of the insights of these investigations may be extrapolated to our population, although this topic has not specifically been addressed in any local investigation.

One limitation of our study is that the study period is relatively short, particularly for the incidence analysis. Because CRC is uncommon among young adults, the number of cases is relatively small and, therefore, a long period of observation would be ideal to provide robust estimates i.e., significant changes in time trends. In addition, we did not perform analysis by cancer stage, which would probably yield interesting results to further explain our findings by age group and the role of screening.

## 5. Conclusions

Our research confirms an increase in CRC incidence rates among young people, and unveils sex disparities in incidence and mortality trends among the older population, with more favorable results among females. These findings may represent a valuable input for health authorities for the implementation of prevention and screening programs, and to estimate the need for relevant health services to tackle the growing burden in specific age groups or population subgroups. CRC cancer prevention, early diagnosis, treatment, and survivorship policies in general do not consider sex-specific differences. Greater insight into how biological and cultural aspects affect the CRC burden by sex may, therefore, lead to more efficient strategies. Tailored policies addressed to the younger population and a gender perspective for the elderly must be considered when prevention and promotion actions are planned and implemented.

## Figures and Tables

**Figure 1 ijerph-18-08232-f001:**
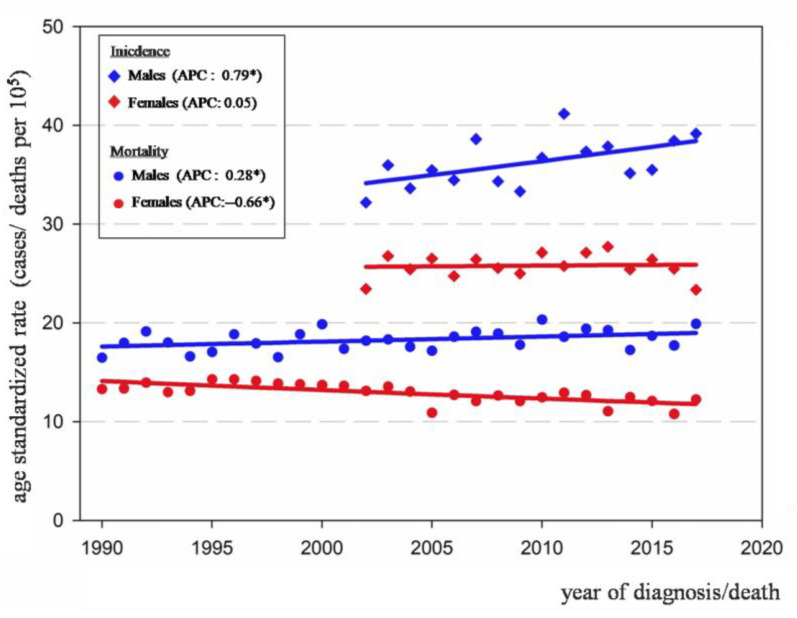
Trends of colorectal cancer incidence 2002–2017 (in diamonds) and mortality 1990–2017 (in circles), all ages by sex, Uruguay; (* *p* < 0.05).

**Figure 2 ijerph-18-08232-f002:**
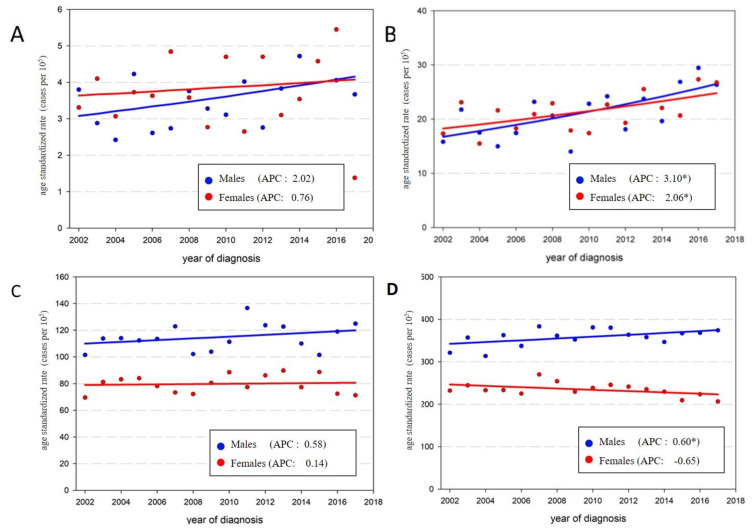
Age-specific colorectal cancer incidence and annual percentage change (APC) by sex and age groups, Uruguay, 2002–2017. (**A**) 20 to 39 years, (**B**) 40–49 years, (**C**) 50–69 years, (**D**) 70 years and older; (* *p* < 0.05).

**Table 1 ijerph-18-08232-t001:** Number and percentage of incident colorectal cancers by age group and sex, Uruguay, 2002–2017.

Age Group	Male		Female		Both	
	Cases	%	Cases	%	Cases	%
All Ages	13,791	100	13,770	100	27,561	100
20–39	270	1.96	297	2.15	567	2.06
40–49	650	4.71	700	5.08	1350	4.90
50–69	5480	39.73	4433	32.19	9913	35.97
70+	7376	53.48	8326	60.46	15702	56.97

**Table 2 ijerph-18-08232-t002:** Colorectal cancer incidence age standardized rates per 100,000 (ASR); annual percent change (APC) and 95% confidence interval (CI) by age group and sex, Uruguay, 2002–2017.

Age Group	Male				Female			
	ASR 2002	ASR 2017	APC	CI 95%	ASR 2002	ASR 2017	APC	CI 95%
All Ages	32.19	39.15	0.79 *	(0.12; 1.47)	23.43	23.35	0.05	(−0.53; 0.64)
20–39	3.80	3.67	2.02	(−0.10; 4.19)	3.31	1.38	0.76	(−2.52; 4.15)
40–49	15.81	26.34	3.10 *	(1.21; 5.03)	17.31	26.74	2.06 *	(0.49; 3.66)
50–69	101.5	124.97	0.58	(−0.42; 1.59)	69.63	71.34	0.14	(−0.88; 1.18)
70+	321.16	374.06	0.60 *	(0.02; 1.20)	232.19	206.72	−0.65	(−1.37; 0.07)

ASR: Age Standardized Rates; APC: Annual percent change; * *p* < 0.05.

**Table 3 ijerph-18-08232-t003:** Colorectal cancer mortality age standardized rates per 100,000 (ASR); annual percent change (APC) and 95% confidence interval (CI) by age group and sex, Uruguay, 1990–2017.

Age Group	Male				Female			
	ASR 2002	ASR 2017	APC	CI 95%	ASR 2002	ASR 2017	APC	CI 95%
All Ages	16.48	19.89	0.28 *	(0.02; 0.53)	13.29	12.25	−0.66 *	(−0.92; −0.4)
20–39	1.14	1.44	1.23	(−0.55; 3.04)	0.84	1.03	−0.61	(−2.75; 1.57)
40–49	8.36	7.74	−0.43	(−1.38; 0.53)	8.01	8.07	−0.59	(−1.59; 0.42)
50–69	48.01	52.56	−0.25	(−0.72; 0.22)	36.60	34.88	−0.61 *	(−1.07; −0.14)
70+	187.03	253.61	0.74 *	(0.47; 1.01)	156.00	135.24	−0.68 *	(−1.02; −0.34)

* *p* < 0.05.

## Data Availability

Data available on request due to personal data protection restrictions.

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
