# Peer review of "Colorectal Cancer in Young and Older Adults in Uruguay: Changes in Recent Incidence and Mortality Trends"

_ijerph, 2021, doi:10.3390/ijerph18158232_

Round 1

Reviewer 1 Report

In the current research article, the authors analyzed trend changes of incidences and mortalities of colorectal cancer in young and older adults in Uruguay. Their data are from Uruguay Cancer Registry. The study is well designed and performed. The authors also concluded some meaningful summaries. I only have some very minor comments:

  1. The English of this article might need further improved.
  2. Some of the figures need to be in high resolution.

Author Response

Thank you very much for your comments.

We improved the quality of the figures and tables.

A native speaker provided some language corrections.

Reviewer 2 Report

The work represents importance for the study area, however, the behavior varies from geographic area to another.

It is necessary to highlight and reinforce the importance of the project for the publication of an international character and that the governmental systems no longer contribute as part of their international obligations.

Author Response

Dear Reviewer

Thank you very much for your thoughful comments.

We agree that the geographical differences were not sufficiently highlighted in the original version.

We added some information to explain those differences.

We also intended to reinforce the importance of sex differences for policies planning.

We hope this new version of the manuscript properly reflects the corrections that you
kindly suggested.

Best Regards,

Carina Musetti

Reviewer 3 Report

An article "Colorectal cancer in young and older in Uruguay: Changes in recent incidence and mortality trends" ia s good data analysis of incidence of colorectal cancer in this country. Attached figures and tables acquire data easily. This article should be using for promotion of healthy lifestyle behavior.

Author Response

Dear Reviewer,

Thank you for your kind remarks, that we greatly appreciate.

Some improvements were added to the original manuscript, according to the reviewers suggestions.

We hope that you will find this new version interesting.

Best Regards,

Carina Musetti